# How Could Sensor-Based Measurement of Physical Activity Be Used in Cardiovascular Healthcare?

**DOI:** 10.3390/s23198154

**Published:** 2023-09-28

**Authors:** Megan E. Hughes, Timothy J. A. Chico

**Affiliations:** 1Clinical Medicine, School of Medicine and Population Health, The Medical School, University of Sheffield, Beech Hill Road, Sheffield S10 2RX, UK; 2British Heart Foundation Data Science Centre, Health Data Research, London WC1E 6BP, UK

**Keywords:** sensor based, activity measurement, cardiovascular healthcare

## Abstract

Physical activity and cardiovascular disease (CVD) are intimately linked. Low levels of physical activity increase the risk of CVDs, including myocardial infarction and stroke. Conversely, when CVD develops, it often reduces the ability to be physically active. Despite these largely understood relationships, the objective measurement of physical activity is rarely performed in routine healthcare. The ability to use sensor-based approaches to accurately measure aspects of physical activity has the potential to improve many aspects of cardiovascular healthcare across the spectrum of healthcare, from prediction, prevention, diagnosis, and treatment to disease monitoring. This review discusses the potential of sensor-based measurement of physical activity to augment current cardiovascular healthcare. We highlight many factors that should be considered to maximise the benefit and reduce the risks of such an approach. Because the widespread use of such devices in society is already a reality, it is important that scientists, clinicians, and healthcare providers are aware of these considerations.

## 1. Introduction

### 1.1. Cardiovascular Disease and Its Global Impacts

Cardiovascular disease (CVD) incurs a substantial burden of long-term disability and causes around 17.9 million deaths per year [1]. Many factors drive the development of CVD, including genetics, demographics, and societal and environmental influences [2]. Despite significant advances in public health (such as improved air quality and reduced rates of smoking) alongside an increasing range of medical and interventional therapies that can prevent or treat CVD, previously declining trends in deaths from CVD are beginning to plateau in higher-income countries [3]. Furthermore, low- and middle-income countries with previously falling rates of CVD have started to show increases in CVD, driven in part by the adoption of more “Western” lifestyles, such as lower levels of physical activity [4,5]. 

In the United Kingdom, 120,000 over 75-year-olds die of CVD each year, accounting for three-quarters of cardiovascular deaths in the UK [6]. One-quarter of all deaths are caused by CVD, representing the biggest cause of premature deaths in economically deprived areas [7].

In addition to death and disability, CVD consumes a huge amount of healthcare and societal economic resources. CVD costs the UK GBP 15.8 billion a year in total, with GBP 7.4 billion spent on healthcare and 6.8 million people living with these conditions, which reduces the ability to work and be economically active [1]. 

### 1.2. Definition of Physical Activity and How This Is Currently Assessed in Healthcare

Physical activity is defined as any movement by the skeletal muscles that uses energy [8]. Two key components of physical activity are heart rate and energy expenditure. The heart rate reflects the intensity of physical activity, whilst energy expenditure helps regulate body mass or composition [9].

During current routine healthcare encounters (such as clinic visits), a person’s usual level of physical activity is almost always assessed using a subjective self-reported description. Such self-report provides no objective measurement and is highly inaccurate [10]. It is difficult for patients to accurately recall and describe their activity levels in day-to-day life. This may be particularly difficult for patients who speak a different language than the clinician, have dementia, or other conditions that prevent communication or recall. Important aspects of physical activity that the current reliance on self-report is prone to miss include information on the intensity of activity, distance covered, and time spent being sedentary [11]. Although several structured questionnaire-based approaches capture self-reported activity, these are inaccurate; for example, individuals are poor at assessing light and intermittent short bursts of vigorous physical activity [12]. Despite the existence of such structured questionnaires, they are rarely used in clinical care, with each clinician instead asking the patient different questions in different ways in an attempt to judge what activities a person can and cannot perform.

Despite the obvious limitations of self-reported data, clinicians rely heavily on them to categorise and estimate risk in patients with CVD. For patients with heart failure, the most used classification is the New York Heart Association (NYHA) Functional Classification. This places a patient in one of four categories based on how limited a patient is by breathlessness during physical activity (1 being no limitation and 4 representing limitation even at rest) [13]. The Canadian Cardiovascular Score used for angina is similar, with 4 grades based on the degree of activity that induces symptoms of angina [14]. Decisions on treatment (such as whether to alter medication or perform surgery) are often based on such categorisation despite their subjective and inaccurate nature.

Objective measurement of a person’s ability to be active is rarely performed due to time and other constraints. A 6 min walk test is sometimes used to assess response to interventions in patients with heart or lung disease. The distance a person can complete in six minutes is correlated with morbidity and mortality and their ability to perform daily activities [15]. However, performance on the artificial six-minute walk test reflects what someone is physically capable of, rather than what their level of activity in daily life is [16] and so does not allow quantification of activity in daily life.

### 1.3. The Relationship between Physical Activity and CVD

Although CVD has many different causes, around 70% of cases are due to a limited number of factors that are potentially modifiable by altering an individual’s behaviour [17]. Declining rates of smoking mean that increasing physical activity is rapidly becoming the most effective way to reduce the risk of many future diseases. Physical activity is an independent risk factor for CVD [18], with low levels of physical activity increasing the risk of both CVD and all-cause mortality [11]. A total of 28% of over-18s globally do not meet guideline-recommended levels of physical activity [19]. Risk scores are used to estimate a patient’s 10-year risk of developing CVD, but these do not consider changes in individualised risk, which includes lifestyle habits [11]. 

The association of physical activity with the risk of both all-cause mortality and CVD persists even when corrected for age, gender, and weight [20]. An increase in 1000 steps a day over a person’s current level decreases the risk of CVD by an estimated 0.18% [12]. Therefore, there are still health benefits from increased activity, even to low levels [20], compared with a sedentary lifestyle [21]. 

The association between physical activity and reduced risk of CVD is apparent at all intensities. Total, light, moderate, intensive, and very short bursts of vigorous activity are all associated with lower risk of CVD and all-cause mortality [12]. At least 150 min of moderate to vigorous physical activity each week is recommended in various guidelines for adults to reduce the risk of CVD [8,12]. It is not only how often someone is physically active, but a person’s overall physical fitness that is important for CVD risk and longevity [5].

## 2. What Digital Technologies Could Measure Activity and What Work in This Area Has Been Performed?

A total of 25% of US adults use a wearable device that is able to measure physical activity, and this proportion is increasing [22]. By 2025, wearable device sales are predicted to reach 520 million units worldwide [23]. In addition to measuring activity, such devices and their accompanying software often provide prompts intended to increase levels of activity, such as goal setting, reminders, and mechanisms for positive feedback [24]. Although wearable devices are better at capturing activity continuously, smartphones are also able to measure activity (while they are carried) and are used by over 90% of the US population [25,26].

### 2.1. Aspects of Healthcare That Could Be Enhanced by the Sensor-Based Measurement of Activity

Healthcare interventions aim to improve a person’s future outcomes at three discrete points of disease development (Figure 1). The earliest is the stage of *prevention*; aiming to reduce the chance of a currently healthy individual developing disease in later life. The next stage is diagnosis; attempting to understand whether a person has an underlying disease, usually because they have developed symptoms that might suggest disease.

The symptoms of possible cardiovascular disease are very common (such as chest discomfort, breathlessness on exertion, or dizziness) and often do not reflect heart disease; instead, they are caused by other medical issues or no significant disease. This means that the great majority of diagnostic assessments lead to the exclusion of heart disease but consume a significant amount of time and healthcare resources. When diagnostic processes detect underlying cardiovascular disease, the final phase of healthcare is treatment, which attempts to improve the impacts of the disease on lifespan (prognosis) and quality of life (such as the ability to work, be active, or other aspects of health). There is a strong rationale to believe that the sensor-based measurement of activity could enhance many aspects of prevention, diagnosis, and treatment [27,28,29], which we describe below.

### 2.2. The Sensor-Based Measurement of Activity for the Prediction of Future CVD

The relationship between the amount of physical activity and future risk of developing CVD (primarily myocardial infarction and stroke) has been apparent for many years and is based on large epidemiological studies that used self-reported levels of activity from questionnaires [30].

Although epidemiological studies using self-report successfully detect associations between activity and risk of disease, this method of assessing activity is inevitably subjective and error-prone, requiring a large study size to detect the relationship between activity and risk of CVD. Increasing evidence now shows that using sensor-based measures of activity provides a more accurate assessment of the “dose-response” relationship between activity and risk of CVD and detects more subtle aspects not detected using questionnaires. Daily physical activity measurement using a smartwatch found that higher levels of activity are associated with a lower risk of CVD [12,31,32], and the ability of wearables to detect very short bursts of vigorous activity (such as going upstairs) reveals that even in people who perform low levels of activity, such bursts are associated with a reduced risk of future CVD [33]. This raises the possibility that the incorporation of the sensor-based measurement of activity into current risk prediction approaches (that use factors such as smoking, blood pressure, and cholesterol levels) may improve the ability to identify an individual’s risk of future MI and stroke or other forms of CVD (such as atrial fibrillation or heart failure) where there are few well-established risk prediction tools. The ability to examine the relationship between both total activity and activity patterns (such as seasonal change, trajectories over time, and intensity of activity) may also provide novel risk factors for incorporation into predictive medicine. Although the accurate prediction of risk is important, it does not improve outcomes unless it leads to a change in behaviour, treatment, or other factors that reduce this risk. The ability to better target support to increase physical activity, or the use of medications such as cholesterol-lowering agents (such as statins) or blood pressure-lowering medications, could lead to more cost-effective healthcare by reducing the proportion of people taking such drugs unnecessarily and increasing the ability to identify people in need of such medications who are currently deemed low risk.

### 2.3. Sensor-Based Measurement of Activity for the Diagnosis of Current CVD

Many forms of CVD cause the person to experience symptoms when active, such as angina pectoris (which causes exertional chest discomfort or pain) and heart failure (which causes breathlessness on exertion). Although there is a range of investigations available to determine whether such diseases are present, almost all require access to specialist centres and incur substantial costs [34,35,36,37].

This means the majority of people with such symptoms are often not offered such investigations, particularly when the assessing physician considers that the risk of underlying CVD is low because of the person’s demographic or other characteristics (for example, in young people where the prevalence of CVD is low), or because the self-reported description of symptoms and limitation of activity is less typical of CVD. This reliance on the self-reported description of symptoms to decide whether or not to refer for specialist investigation suffers the same flaws as the reliance on self-reported physical activity described above.

The ability to obtain objective measurements of physical activity could benefit the initial diagnostic assessment in several ways. For conditions such as angina that are typically induced on exertion, the ability to “label” episodes with the level of activity at the time (at rest, recent exertion, exertion, etc.) would provide contemporaneous data to inform the context of the symptoms. An objective measurement of the level of activity and how this has changed over time, particularly from a previous baseline of “good” health, would allow the physician to judge quantitatively the impact of the possibly underlying disease and provide the ability to identify and prioritise those with the most significant limitation for specialist investigation. This would be aided by the ability to compare an individual’s level of activity not just to their own previous baseline but to normative, age-adjusted ranges (similar to how many blood test results are reported and interpreted).

### 2.4. The Sensor-Based Measurement of Activity for Monitoring Treatment of CVD

Because many forms of CVD limit a person’s ability to be active (such as by causing chest discomfort or breathlessness on exertion), treatments including pharmacological therapy or surgical interventions, such as stents or valve replacement, aim to improve these limitations. Because most of the clinical evidence for such treatments arises from before the availability of smartphones and wearables able to measure activity, there is little evidence that shows the magnitude of any improvements, and whether certain subgroups of patients experience greater or less benefit.

Two key types of clinical studies provide evidence for clinical decision making: randomised clinical trials (RCTs) and clinical registries. In RCTs, the effectiveness and safety of a novel treatment or intervention is examined by randomly allocating patients to receive either the novel treatment or a comparator, which is usually the previous standard of care [38,39].

RCTs provide the “gold-standard” of clinical evidence as they overcome bias, such as in drug trials, where neither the patient nor the clinician knows whether the patient is receiving the active medication or an inactive placebo. Surgical and interventional trials are more challenging because of the difficulty in such “blinding” to treatment allocation, although recent innovative studies have shown that it is possible to perform sham interventions on patients who are unaware whether they have received an intervention or sham operation [40].

Clinical registries are observational studies that recruit cohorts of people with a specific disease or after a specific treatment or exposure [39]. These usually follow patients over time to understand the trajectories of disease, response to treatment (in a non-randomised manner), and other outcomes [41].

Both RCTs and clinical registries are beginning to incorporate measurement of activity by sensor-based technologies [42,43]. This will accumulate evidence that would allow patients and physicians to predict the effect of different treatments on the person’s ability to be active and may also identify those people who gain the most and least from such treatments.

### 2.5. How Should Activity Be Measured in Healthcare?

A range of approaches and devices have the potential to measure a person’s activities. The more sensors used, the better the estimate of physical activity and energy expended [11]. The most basic technologies are pedometers, which are low cost and provide continuous, passive measurement of activity while worn [24]. There are concerns with pedometer accuracy, reliability, and validity [24] that make other devices, primarily wrist-worn wearables, more likely candidates for adoption in healthcare. Such wearables predominantly use triaxial accelerometers that measure the intensity and frequency of linear acceleration along three orthogonal planes [11,24]. The raw accelerometer data are then processed to attempt to identify aspects of physical activity, particularly rhythmic accelerations and decelerations associated with taking a series of steps. The ability to reliably identify steps taken by the wearer allows calculation of other derived metrics such as sedentary time, light, and moderate and vigorous-intensity activity. Moderate-to-vigorous activity is associated with lower mortality compared to sedentary behaviour or light physical activity [11]. Such devices may be less able to capture very short episodes of activity, such as going from sitting to standing [44]. Some wearables use differential capacitive accelerometers. These benefit from low power usage, fast response to motion, and better accuracy compared to piezoresistive accelerometers, but are more sensitive to temperature [11].

The addition of a GPS (to identify location) and a barometer (to measure altitude) may increase the accuracy of assessment of outdoor activity, although barometer accuracy can be affected by both temperature and atmospheric pressure changes [11].

Because mobile phones now often include accelerometers and GPS location functions, they are also able to assess physical activity when carried. The accuracy of the identification and measurement of steps is lower than wearables [26,45], and the tendency to not carry phones when at home reduces the ability to measure activities, such as stair climbing, and the ability to discriminate sedentary time, both of which are associated with CVD risk. Conversely, phones have the advantage of being able to obtain other health-relevant data, particularly user-generated data on symptoms, quality of life, and other measures necessary to understand the user’s health and the effects of disease and treatment [45,46].

### 2.6. Can the Measurement of Physical Activity Help Increase Levels of Physical Activity?

Although the measurement of physical activity is highly likely to provide the user or clinician with medically useful information, it remains less clear to what extent the availability of such data increases an individual’s levels of physical activity.

Although most treatments for CVD are either pharmacological or surgical, physical activity is also a treatment for CVD. This is well demonstrated by cardiac rehabilitation (CR) programmes after a heart attack (myocardial infarction, MI). These multi-faceted programmes educate patients about their condition, its treatment, and behavioural modifications that reduce the risk of future MI. A central component is the promotion of and support in increasing levels of physical activity, and this has been shown to halve the chance of future adverse cardiac events. Although traditional cardiac rehab programmes have used face-to-face group and individual sessions, these are less accessible for many types of people (such as those in work), and uptake of CR remains around only half of the eligible population, with an even smaller proportion completing the entire programme [47,48].

The ability to measure and monitor physical activity outside of direct observation by a clinical care team provides the ability to deliver CR either partially or fully remotely without the need for in-person attendance. Home-based cardiac rehabilitation facilitated by mobile technologies, including a wide range of activity monitors, has shown similar outcomes to conventionally delivered CR [49].

Formal CR programmes, even if delivered remotely, are resource intensive. There is an urgent need for approaches that increase physical activity in much larger populations than CR can be delivered to, to reduce the burden of diseases such as obesity, type II diabetes, and cancer, which are associated with low levels of physical activity. This has led to considerable interest in using wearables and mobile phones to not just record activity but deliver interventions aimed at increasing physical activity via a range of motivational and gamification approaches [47,48,50]. There is consistent positive evidence that mobile phone digital interventions can increase activity compared with non-active and active controls [51]. Thus, wearables and mobile devices have the potential to both measure and increase physical activity and improve health [5]. The use of wearables improves the number of daily steps and amount of physical activity [52]. Increased physical activity can contribute to weight loss after a period of time [53]. Good adherence to activity tracking shows a positive association with weight loss. This possible temporal relationship could be a good predictor of weight fluctuations. This correlation could be due to a person deciding to engage in behaviour that improves their health, such as healthy eating, which may increase the likelihood of weight loss [54]. Even so, wearable technologies could be used as part of a weight loss programme to encourage physical activity. Healthy lifestyles could be encouraged through the use of interventions using wearables focused on weight and physical activity [55].

Wearable devices have a role to play in motivating users to increase their physical activity, but it is unclear what the benefits are for patients with established chronic diseases [52]. Wearable devices in combination with educational support (e.g., goal setting) may be beneficial for diabetes mellitus in helping to reduce weight and HbA1c compared to standard care [56]. Overall, there still needs to be more evidence to prove the efficacy of wearables to improve outcomes in people with chronic diseases [56,57], as there are insufficient data showing consistent health benefits.

The short-term use of accelerometers may not lead to long-term changes in behaviour [12]. Continued use of behavioural interventions is sometimes difficult, with people only using the digital health intervention for a few days to weeks. This makes it difficult to make the behaviour change consistent [58]. Although digital interventions may be effective for 3–6 months, after 12 months there is little effect [51]. There is insufficient evidence for long-term benefits from digital technologies that seek to increase physical activity. Even though there may only be a small effect with digital interventions, it is generally better than no intervention [51]. To maintain physical activity participation over a longer period, a self-monitoring device could be used in combination with regular contact with a health professional [59]. Dropout rates with these approaches could be due to lack of acceptance, usability, and the user-friendliness of the technology [60].

## 3. Limitations and Other Considerations for the Use of Digital Technologies in Healthcare

### 3.1. Patient Factors

The large-scale deployment of digital interventions may be restricted by patient-related barriers such as health status, user characteristics, privacy issues, security concerns, quality concerns, a lack of personal motivation, and accessibility to digital resources [4].

For digital technologies to obtain data and deliver effective interventions, individuals need to be willing to use them. Some people are hesitant to use technology and healthcare systems, and time needs to be taken to understand their concerns. To build trust, patients as well as healthcare professionals need to be involved in changes to how healthcare will be delivered [61]. Risk perception increases motivation to change behaviour, which could lead to improved clinical outcomes [52], making it important to provide individualised risk assessment and estimation of the effect of increasing activity.

A follow up over a long period of time for self-administered surveys on mobile phones is relatively unexplored. Reduced adherence to completing studies is variable and depends on study participants and the nature and implementation of the survey [62].

Another factor to consider is technology literacy. Groups with known lower digital health usage are older people and those of lower socioeconomic and health status and lower health literacy [4]. However, the increasing ubiquity of these devices may improve health inequities in the longer term. Access to mobile phones is as high in ethnic minority groups compared to the rest of the population in higher-income countries, while the use of mobile phones is steadily increasing in older populations [18]. Even though many older patients use IT, they are still faced with difficulties in accessing and using mobile health technologies. Providing support and training could increase acceptability [63].

Owners of activity monitors have suggested that they have a positive experience when wearing the devices. There was found to be little risk of negative mental health outcomes simply from using the devices [64]. Technology such as smartphones has the potential to be used in the assessment of mental health conditions such as depression, providing practitioners with continuous data on a person’s behavioural patterns that may correlate with mood or other indicators of mental health [65]. 

### 3.2. Accuracy, Reliability, and Validity of Digital Technologies

Differences in device technology and software must be considered. Devices with the highest validity also have the smallest error rates [9]. Wearable devices have been found to generally accurately measure steps in day-to-day life, even though they may overestimate total steps taken. When using these devices to measure moderate to vigorous intensity physical activity, the validity is low [66,67]. The measurement of heart rate, steps, and distance is increasingly reliable with consumer devices, but the measurement of energy consumed is inaccurate [42]. The inconsistencies in the measurements of the different devices could be due to variations in factors, such as sensitivity thresholds and user methods of wearing or using the devices [42,68]. 

The location of the device on the body can affect results and is an important factor in accelerometer accuracy [11]. Placing the sensor centrally on the torso has the least errors compared to other body locations [11,69] when measuring posture, acceleration, and whole-body movements. Generally, placing the accelerometer on the hip is better compared to the wrist when measuring energy expenditure and exercise intensity [69]. A sensor located on the ankle is preferable for the measurement of steps and energy used compared to other areas of the body, such as the trunk and wrist. However, the accuracy of the location must be balanced against compliance and convenience. Wrist placement seems likely to be the best option given the cultural acceptance of wrist-worn jewelry and timepieces [11]. 

Independent validation of devices is important to ensure they are tested for accuracy. For example, there are blood pressure monitors on the market that have not undergone validation testing but have been allowed to go to market by regulatory bodies. This is due to regulatory authorities focusing mainly on ensuring that the devices are safe, but to be able to have confidence in the recordings of the devices, device accuracy needs to be known [70]. 

If activity is going to be tracked using these activity monitors as part of a lifestyle intervention, then the sensitivity and specificity of such monitors are important. The activity monitors tested had good sensitivity for step detection over different surfaces and wearing different types of footwear. However, specificity accuracy is equally important, as devices need to be able to exclude when a movement is not a step and, therefore, disregard it. Currently, there is variation across devices in the detection of steps; this appears to be the least specific when carrying out activities that are non-stepping. Ultimately, the desired activity monitor should be able to differentiate between when a step is being taken compared to non-stepping movements and, therefore, only record the steps that a person has taken [71,72].

There needs to be a universal definition for what is classified as “engagement” in digital health interventions to define the amount of time needed for the person to be engaged to produce changes in clinical outcomes [58].

Ultimately, devices need to be user-friendly for the patient to engage with measurement and any intervention. The mobile app rating scale (MARS) is a tool used to assess mobile apps’ quality and functionality. The MARS consists of a set of criteria to gain an overall score for an app, and this includes several categories, such as functionality and engagement. Consistently, the highest-scoring domain using the MARS scores is functionality, with aesthetics second [62].

Currently, there are many challenges faced when trying to implement wearables into clinical practice, such as practicalities of use and legal and regulatory obligations. To be able to better understand the use of wearable devices and how they can be implemented into clinical practice, it is necessary to conduct clinical trials examining the effect of the addition of such devices and the data they generate to the standard of care, ideally in a randomised clinical trial (RCT). The data from such clinical trials could also facilitate the creation of standards for the evolving industry of healthcare-focused activity monitoring.

### 3.3. Barriers, Challenges, and Possible Solutions

The adoption of new technologies in healthcare always faces a wide range of barriers that need to be addressed. We describe these below, and they are summarised in Figure 2.

### 3.4. Regulation

Healthcare is regulated heavily for good reasons, as new technologies can pose significant risks. However, the current regulatory framework does not cope well with rapidly evolving technology [73]. There is a lack of specific legislation for mHealth apps [4]. Current medical device legislation is not specific to novel technologies, such as wearables. Different criteria need to be tested, verified, and used to inform regulatory decisions [23]. The UK National Institute for Health and Care Excellence (NICE) has started to produce guidance in this area over the last few years [74,75]. EU bodies and the FDA are applying higher scrutiny, with legislation for medical device software extended to apps [4]. EU medical device regulation includes the use of mobile and wearable devices in prediction and prognosis. This includes physiological data, which is collected and evaluated and covers digital health apps with these functions. Other regulatory frameworks include the EU and UK General Data Protection Regulation, which covers all personal data [23].

The FDA is amending some regulatory approaches to better keep up with rapidly evolving technologies. This may enable new technologies to be efficiently developed and tested whilst ensuring they are safe and effective. The FDA is in the process of creating a digital health innovation plan. In the US, the 21st Century Cures Act has been created. Not all mobile health technology devices need approval by the FDA, such as apps with the intended use to promote a healthy lifestyle without a clinical application, as these are deemed to be low [76]. FDA regulations will apply to technologies that could potentially put the patient’s safety at risk if the technology does not function properly [77]. In the UK, if a consumer is using an app for medical reasons, it requires a UKCA or CE mark, confirming that the app or device is safe to use and fit for the intended clinical use [78].

### 3.5. Privacy, Data Exploitation, and Surveillance Concerns

In one study of the 600 most used mobile health apps, 69.5% did not have a privacy policy [79]. Apps often gather sensitive information, which may then be shared with third parties [23]. Metadata associated with a user could be used to re-identify them [11]. Patients should provide explicit consent to allow their data to be used in research or shared with third parties [11]. Cybersecurity breaches are always possible, especially when platforms are not secure [11].

Despite these considerations, 81.9% of US adults were willing to share data from wearables with their healthcare team, with 69.5% willing to share with family and friends [22]. In determining a patient’s willingness to share data with providers, there are several key factors. These include health self-efficacy, trust in providers as an information source, frequency of wearable use, current use of other mHealth technologies to help communicate with providers, and levels of physical activity [22].

To reduce the amount of raw data being transferred and reduce the impact of security breaches, on-device data analysis may be preferable, although this consumes battery and computational power [44].

Data breaches of some form are an inevitability due to the movement of data across platforms [11]. For stakeholders, the supervision of medical data storage is essential [44]. The use of cloud services is becoming increasingly popular; however, it is often unclear where the data are stored, which could lead to a breach of some privacy rules, such as which country data is transferred to. Therefore, the data may not only be at risk during transmission to storage but there may be another security threat if they are stored in an unknown location. An ideal solution would be to have the data stored in a secure room onsite, with restricted access [80] within hospital networks [44]. Blockchain systems are increasingly being developed to store digital data [81,82]. 

Wearables and other medical devices are a developing industry that needs standards to ensure the security of data [83]. IEEE has started to produce guidelines for these emerging devices (IEEE standard 360-2022). It outlines specifications for the technical needs and methods of testing wearables. This ranges from suitability for wearing the device to functionality. The use of the IEEE standards in device design is voluntary. However, without the standardisation of devices, data quality and reliability are affected [84]. 

### 3.6. Data Synthesis

The volume and velocity of the data generated with the near-continuous measurement of physical activity pose challenges. The data are often lacking in generalisability and validation. Such large amounts of data are impossible to review within the short amount of time clinicians have to see an individual patient [4,85]. There are concerns as to whether doctors would be liable for not reviewing data transmitted outside of office hours [4]. Actively reviewing incoming data and providing feedback to patients would require a large amount of resources and time if performed by clinicians [4]. Software tools are needed that summarise physical activity data and provide these to the user and clinician, using an intelligent filter. This needs to be reliable, clinically relevant, documented in the electronic health record, and easy to review, similar to lab or imaging results. Wearable device data ideally needs to be combined with clinical data and symptom information into one screen so that the electronic health record is updated with information in real or near real-time [5,85,86]. 

### 3.7. Health Inequalities and Cost

There is a strong link between socioeconomic status and health literacy and socioeconomic predictors of e-health use among adults who use the internet. Interventions for measuring physical activity digitally do not have the same outcomes for lower socioeconomic groups compared to higher [87]. Lower levels of education indicate a lower likelihood of searching online for a healthcare provider [88] or using the internet or email to engage with doctors. The strongest predictor of low health literacy was financial deprivation [4]. A survey reported a 3-fold difference among 4272 US adults using a wearable device between participants who earned USD >75,000 compared to those who were earning USD <30,000 [11]. People in more deprived areas perform less physical activity [89], and there is little evidence to show that physical activity interventions in lower socioeconomic groups are effective. The type of behaviour change technique used also has no effect [87].

The number of behaviour change techniques incorporated into a device is not necessarily reflected in the device price [90]. There was no evidence for the cost-effectiveness of complex smartphone communication as an intervention. There is some evidence that text-based communication can be cost effective [91]. Compared to comparative methods, mHealth interventions are not significantly different when looking at costs and benefits. Evidence is limited in mHealth interventions for the overall cost-effectiveness of the economic evaluations [91].

Access to digital health is not universal due to cost or availability. Individuals may not have access to devices, such as mobile phones. Other barriers include a lack of internet access or an adequate data plan or network. Devices and operating systems often need to be updated on a frequent basis, which could lead to compatibility issues [50]. 

### 3.8. Implementation

A major challenge is the limited application of physical activity measurement within the clinical workflow [5]. Digital healthcare is currently used as an add-on to existing care rather than being integrated into existing care pathways. The use of digital technologies would require a significant redesign of clinical pathways, and mHealth needs to be compatible with hospital IT systems [4,73]. The use of wearables in clinical practice needs further validation in both patient and healthy populations. Accelerometers have not been adequately tested in daily activities in which there is no walking involved. Validation of these devices may struggle to keep pace with advances in technology. The consistent use of wearables over the long term (for example, years) has not been proven to be feasible, particularly if users change devices [24].

## 4. Conclusions

The sensor-based measurement of physical activity has the potential to be extremely valuable for many aspects of cardiovascular healthcare. Many users already use sensor-based devices to track their activity with the expectation that these data will be useful to them and their clinicians to understand their health and enhance their healthcare. However, although simple in principle, we have described many complex considerations that need to be taken into account in planning the future integration of such technologies in healthcare. This requires urgent attention because the increasing ubiquity and use of these technologies gives us no option other than to ensure they are used in a way that provides the greatest benefit while posing the lowest risk. We, therefore, make the following recommendations. 

Firstly, device manufacturers and regulators should work with clinicians and researchers to agree on the standards of accuracy and validity required for use in healthcare and make such testing results openly available.

Secondly, there is an urgent need for clinical studies that evaluate the clinical impact of providing objectively measured physical activity on patient and clinical outcomes. Such outcomes could include patient satisfaction, clinician confidence, or more conventional outcomes such as health resource usage, hospital admission rates, or even rates of CVD and death.

Thirdly, there is an urgent need for large-scale, long-duration studies that use devices to measure physical activity over years and correlate this with the incidence of CVD and other major diseases. Such studies would demonstrate more clearly the potential benefit of incorporating sensor-based physical activity measurement into clinical risk prediction.

Fourthly, although the measurement of physical activity is a key component of attempts to improve population health by increasing physical activity levels, it must be part of a complex approach that also addresses the deep-rooted cultural, socio-cultural, built environment, and infrastructural barriers to greater levels of physical activity. Without such an approach, technology alone cannot reduce the growing burden of physical-inactivity-related CVD. 

## Figures and Tables

**Figure 1 sensors-23-08154-f001:**
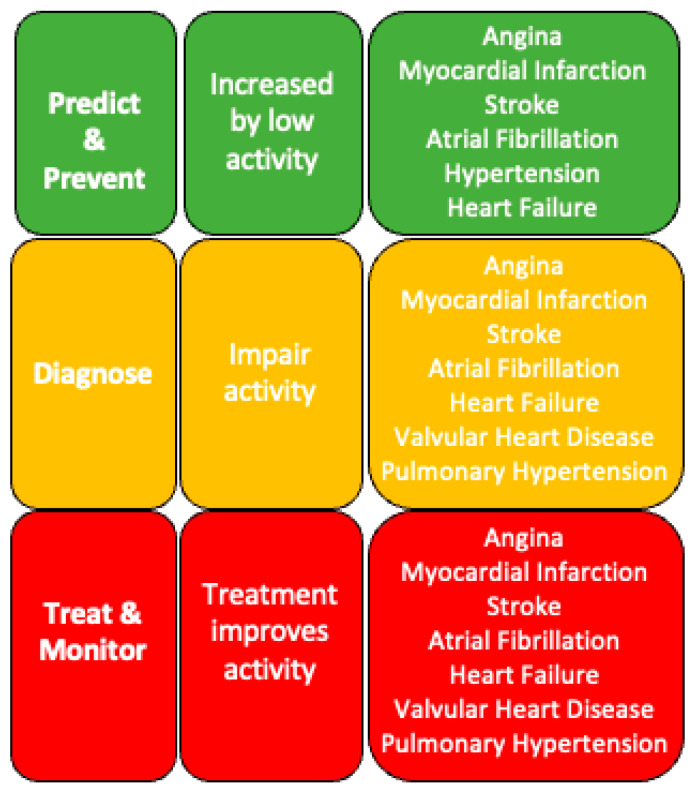
The relationship between physical activity and prediction, prevention, diagnosis, treatment, and monitoring of common cardiovascular diseases.

**Figure 2 sensors-23-08154-f002:**
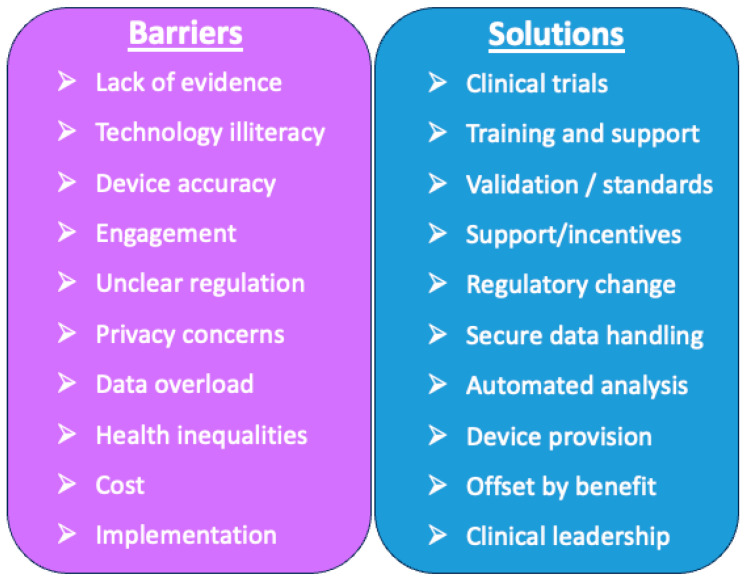
Barriers and possible solutions to adoption of the sensor-based measurement of physical activity in healthcare.

## Data Availability

Not applicable.

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
