# Peer review of "How Could Sensor-Based Measurement of Physical Activity Be Used in Cardiovascular Healthcare?"

_sensors, 2023, doi:10.3390/s23198154_

Round 1

Reviewer 1 Report

Wearable devices measuring the physical activity helps in Cardio vascular healthcare. Authors did a wonderful work with detailed review. However I have found few things to be included in the review manuscript. 

1. Authors are asked to discuss the specificity and sensitivity of the currently used sensors-based wearable devices with examples in separate heading with references.

2. While discussing about wearable devices, Line - 295 to 297 is odd to the flow. Authors are asked to remove the sentences.  

3. Authors are asked to add more specific points on 'Shortfalls of sensors-based wearable devices'. 

4. Authors are asked to discuss the digital devices in connection with mental health of the patients.

Authors are asked to revise the manuscript accordingly and submit.

Author Response

please see the attaachment

Reviewer 2 Report

This a very interesting research, but it should be not a literature review, it will be a systematic review following PRISMA checklist with possible inclussion of meta analysis. It will appear a discussion section before conclussions. Literature reviews ussually are written by recognized experts in the field and actually its use is very small

Reviewer 3 Report

Overall, this highly informative and well-referenced manuscript has significant potential to add the field of sensor research.  The current version is comprehensive and identifies many of the critical issues in the use of wearables in cardiovascular healthcare. There are just a few comments that the authors may want to entertain.

Quite frankly, this is a very well-written and clear manuscript that does a superb job bridging medicine and technology.  The narrative is easy to follow and, indeed, this could easily qualify as a book chapter. There is a lot of information to digest, however, and some sections are quite long. Section 2.2 is approximately 100 lines long, and it may help the reader to have some additional subheadings for reader wayfinding. No specific subheadings are suggested, but the section covers a broad area of tracking activity levels, the distinction between objective and objective measures, and social media (++). Not a huge challenge, but as the end of the section draws to a close, it is hard to recall a specific “take-away”.  At one point I though the authors were going to conclude that wearables are “not accurate for clinical care, but OK for high-level risk management” (but am not sure if they had that in mind).

·       Similar to section 2.2, with section 3.2 in mind, the authors lead up to technology challenges and they were going to mention that it is entirely possible to launch, say, an automated blood pressure machine that does not meet certain standards – that is, if are there any real standards at all? Any comments in that regard would inform the readership. Not that the US Blood Pressure Validated Device Listing (VDL™) is the beacon for design standards (https://www.validatebp.org/) but there are dozens of devices listed and there is no question that some are not clinically qualified.  Lines 381 – 390 offer an opportunity to comment on the need for clinical trials that are used to assess if devices meet standard-of-care.

·         Section 3.3.2 offers the opportunity to comment a bit further on cybersecurity in the context of evolving IEEE Cybersecurity standards. Maybe just mentioning that there are standards, but the lack of use of standards compromises device design in/for clinical settings. The idea is that there is emerging guidance, but there is no requirement designers actually follow for non-medical applications.

·         The conclusion seems to erode a bit of the  enthusiasm that builds while reading the manuscript. It would add so much to the manuscript if some specific and pointed comments were added.

Round 2

Reviewer 2 Report

This litterature review is very exhaustive, but to future researches I suggest perform systematy reviews with meta analysis, because it offers evidence base kwnoledgment